# The Use of Modified Fe_3_O_4_ Particles to Recover Polyphenolic Compounds for the Valorisation of Olive Mill Wastewater from Slovenian Istria

**DOI:** 10.3390/nano12142327

**Published:** 2022-07-06

**Authors:** Kelly Peeters, Ana Miklavčič Višnjevec, Črtomir Tavzes

**Affiliations:** 1InnoRenew CoE, Livade 6a, 6310 Izola, Slovenia; crtomir.tavzes@innorenew.eu; 2Andrej Marušič Institute, University of Primorska, Muzejski Trg 2, 6000 Koper, Slovenia; 3Faculty of Mathematics Natural Sciences and Information Technologies, University of Primorska, Glagoljaška 8, 6000 Koper, Slovenia; ana.miklavcic@famnit.upr.si

**Keywords:** polyphenolic compounds, olive mill waste water, modified iron (II; III) oxide particles, adsorption and desorption, magnetic collection, qualitative and quantitative analysis

## Abstract

Olive mill waste water (OMWW), a by-product created during the processing of olive oil, contains high amounts of polyphenolic compounds. If put to further use, these polyphenolic compounds could be a valuable resource for the speciality chemical industry. In order to achieve this, isolation of the polyphenolic compounds from OMWW is needed. Several techniques for this process already exist, the most widely used of which is adsorption beds. This research describes new ways of collecting polyphenolic compounds by using unmodified iron oxide (Fe_3_O_4_) particles and Fe_3_O_4_ modified with silica gel (Fe_3_O_4_@C18), citric acid (Fe_3_O_4_@CA), and sodium dodecyl sulphate (Fe_3_O_4_@SDS). This approach is superior to adsorption beds since it can be used in a continuous system without clogging, while the nano-sized shapes create a high surface area for adsorption. The results of this study show that, if used in a loop system of several adsorption and desorption cycles, (un)modified Fe_3_O_4_ has the potential to collect high concentrations of polyphenolic compounds. A combination of different modifications of the Fe_3_O_4_ particles is also beneficial, as these combinations can be tailored to allow for the removal of specific polyphenolic compounds.

## 1. Introduction

Olive oil produced by three-phase decanter systems creates two by-products: olive mill wastewater (OMWW) and pomace. The latter contains a much higher share of polyphenolic compounds than the oil does [1]. Our research is focused on the collection of these polyphenolic compounds from OMWW. Due to this high concentration of phenolic compounds, OMWW is considered to be one of the most polluting effluents produced by the agro-food industry [2]. However, polyphenolic compounds are linked to positive effects on human health, and they exhibit antimicrobial and antioxidant properties [3]; as such, this by-product could be effectively put to further industrial use in the cosmetic, pharmaceutical, and food industries. 

Much research has been conducted on the isolation of polyphenols from OMWW. The investigated methods made use of adsorbents [4,5,6,7], ultra- or nano-filtration membranes [8,9,10,11], micro-wave assisted solvent extraction [12], drowning-out crystallization-based separation [13], and co-precipitation reactions [14] to recover polyphenolic compounds. Reviews of the different polyphenol recovery methods can be found in the literature [15,16,17,18,19].

Of these techniques, adsorption is the most widely used and effective technique for removing environmental pollutants [20]. There exist cheap adsorbents, such as activated carbon, coal fly ash, sludge, biomass, and zeolites, but these are not specific and regeneration after chemisorption is not cost effective [21,22]. Meanwhile, other adsorbents are more specific for polyphenols; these are able to adsorb polyphenols from different matrices and can be desorbed in polar solvents. These adsorbents include Amberlite (XAD4, XADHP7, XAD16), sepabeads (SP207, SP700), Isolute C8, Dowex Monosphere 550a, silica particles with polyvinyl alcohol chains modified with N-methyl imidazole proline salt, triamine-grafted mesoporous silicas, and sodium dodecyl sulphate modified alumina [23,24,25,26,27,28,29]. However, this technique comes with certain disadvantages. When using adsorbents, there is uneven saturation of the adsorption bed, and a long processing time is required for the OMWW to run through the whole adsorption bed; moreover, large numbers of adsorption beds are needed. Additionally, fixed beds require total or partial shutdown to replace adsorbents and, if a continuous system is required, there is a need for complex piping and valve arrangements with a control system. Another problem with adsorbents is that they can become clogged with particles, making it necessary to use membranes, which need to be cleaned frequently.

To overcome the disadvantages associated with such conventional techniques, this project focusses on isolating and concentrating polyphenols with the help of iron (II, III) oxide (Fe_3_O_4_) particles modified with a surface coating of adsorbing materials. We aimed to use particles with sizes in the upper nano-scale range. Theoretically, magnetic particles of sizes in the nano-scale have significant benefits. Just 1 g of 10 nm diameter magnetic beads contains as many as 10^18^ particles. This is an incredible amount of potential scavengers for phenols, and it concentrates this otherwise toxic waste into a very small volume. However, working with magnetic particles of such small sizes brings about its own challenges: namely, that beads less than 20 nm in size are fully dispersed in solution [30], making complete magnetic collection much harder than it is with larger particles which tend to agglomerate. Their small size and high redox reactivity also make them potentially harmful to living organisms [31]. The key advantage of using magnetic nanoparticles larger than 20 nm is that they are easily dispersed by stirring or shaking, and easily collected with a magnetic field. Although they are slightly larger than 10 nm particles, they still have a large surface area and therefore they still can adsorb large quantities of chemical compounds. Therefore, they can be deployed into existing technology and infrastructure, and there are few barriers to operational uptake [32].

Until now, research on the modification of magnetic particles for the enhanced extraction of polyphenols was performed by adsorbing ionic liquid or ionic surfactants on the surface of metal oxides [33,34,35,36]. The ionic liquid or surfactant’s hydrocarbon chains provide hydrophobic or π-π stacking interactions for hydrophobic analytes, while the polar groups adsorb ionic analytes via electrostatic interaction or a hydrogen-bonding interaction. This system is suitable for the extraction of phenols from aqueous, rather than oily, matrices. Another technique for selectively binding and recovering a selected polyphenol involves the use of magnetic particles modified with molecularly imprinted polymers. Wang et al. [37] captured hespertin from the dried pericarp of *Citrus reticulata Blanco*, while Ma et al. [38] extracted catechin, epicatechin, and epigallocatechin from black tea. Since these polymers are target specific, they are not useful when several compounds or groups of compounds are wanted (collecting mixtures of compounds can sometimes have beneficial synergistic effects). Ying et al. [39] developed a method that showed the selective attachment of cis-diol polyphenols from fruit juices via columns containing polyethyleneimine modified with 4-formylphenylboronic acid. Gold nanoparticles with a stabilizing layer of cysteamine hydrochloride and 4,4′’-dithiolterphenyl were tested in OMWW and showed promising results [40].

In our investigation, we compared the efficiency with which non-modified Fe_3_O_4_ and several types of modified Fe_3_O_4_ (citric acid (CA), silica gel (C18) and sodium dodecyl sulphate (SDS)) removed polyphenolic compounds from OMWW. The modification agents were chosen not only because they are good adsorbents, but also because subsequent desorption of the polyphenols is possible. Fe_3_O_4_@SDS and Fe_3_O_4_@alumina(Al_2_O_3_)@SDS were chosen according to the research of Adak et al. [23]. Their study stated that surfactant-modified alumina possesses the ability to remove phenols from aquatic environments through a process called adsolubilization. Fe_3_O_4_@C18 was chosen because alkyl-functionalized silicas are used as stationary phases in reversed-phase high-performance liquid chromatography. According to the findings of Ottaviani et al. [41], adsorption and desorption can take place depending on the water–solvent ratio of the environment. In aqueous solutions (e.g., OMWW), C18 chains tend to collapse and fold on the silica surface, trapping the phenol. At a higher solvent concentration (desorption media), the chain layer is assumed to have a relatively ordered structure, leading to the release of the phenols. Fe_3_O_4_@CA was used with the idea that polar interactions would adsorb phenols. Even though methanolic desorption is more effective, ethanol was chosen as the desorbing solvent, because it is a polar solvent, which is not particularly harmful to the environment or to human health [42].

The aim of our study was to use the described innovative, unconventional, low-cost techniques, which may be suitable for industrial use, in order to successfully extract polyphenols from OMWW. To the best of our knowledge, this is a unique study, as it uses this extraction method on OMWW with a unique polyphenol composition involving secoiridoids, which are not found in any edible plants other than olives [43].

## 2. Materials and Methods

### 2.1. Materials and Instrumentation

Extraction solvents: ethanol (EtOH) (Carlo Erba, absolute anhydrous for analysis-reagent grade, Emmendingen, Germany). Reagent used to adapt the pH of OMWW: hydrochloric acid (HCl, 37%) (Honeywell, reagent grade, Charlotte, NC, USA). The OMWW was filtered with 200 nm polyamid (nylon) syringe filters before LC-MS/MS measurements. Iron (II, III) oxide particles (Fe_3_O_4_, 50–100 nm, Sigma Aldrich, St. Louis, MO, USA) were used to collect polyphenolic compounds from OMWW. Reagents used to modify the Fe_3_O_4_: citric acid monohydrate (Fisher Scientific, Loughborough, UK, ≥99.8%), toluene, sodium dodecyl sulphate (≥98.5%, GC, Sigma Aldrich, St. Louis, MO, USA), sodium chloride (Honeywell, reagent grade, ≥98%, Muskegon, MI, USA), sodium acetate (Acros Organics, anhydrous, 97%, Geel, Belgium), acetic acid glacial (LabExpert, 99–110%, p.a, Ljubljana, Slovenia), C18-SiCl_3_ (Sigma Aldrich, ≥95%, GC, Buchs, Switzerland), and aluminium isopropoxide (Sigma Aldrich, ≥98%, St. Louis, MO, USA). Solvents used for LC-MS/MS analysis: acetonitrile, MeOH, and water (Honeywell, LC-MS chromasolv grade). Reagents used for the Folin–Ciocalteu (FC) method: FC reagent (Merck, Darmstadt, Germany), gallic acid (97.5–102.5%, Sigma Aldrich, St. Louis, MO, USA), sodium carbonate (anhydrous for analysis, Merck, Darmstadt, Germany).

High-performance liquid chromatography, coupled with electrospray ionisation and quadrupole time-of-flight mass spectrometry (HPLC-ESI-QTOF-MS, 6530 Agilent Technologies, Santa Clara, CA, USA), was used to qualify and quantify the polyphenolic compounds that were present. The HPLC equipment incorporated a Poroshell 120 column (EC-C18; 2.7 µm; 3.0 × 150 mm). An Epoch Microplate Spectrophotometer (Biotek Instruments, Winooski, VT, USA) was used for the determination of the total phenol content (TPC).

### 2.2. Sample Collection

The samples were collected at the Franka Marzi olive mill (N 45° 30.6588 E 13° 42.2574, Koper, Slovenian Istria). Details about the samples can be found in our previous research [44]. Immediately after sampling, OMWW samples were stored in a freezer (−18 °C). Acidic conditions can considerably increase the fraction of free phenolic compounds in OMWW. However, since these experiments were designed to collect polyphenolic compounds from OMWW on a large scale, the OMWW was not acidified, as recommended by Jerman Klen and Mozetič Vodpivec [45], because this would not be economically feasible. Fe_3_O_4_, Fe_3_O_4_@C18, Fe_3_O_4_@CA, and Fe_3_O_4_@SDS were tested for their extraction efficiencies, each on a different day. Therefore, differences can be found in OMWW composition between experiments. However, the whole sequence for each of the four experiments was made on the same date with the same OMWW.

### 2.3. HPLC-DAD-MS/MS Analysis

HPLC-ESI-Q-TOF-MS analysis, along with compound qualification and quantification, was performed as described in our previous research [44]. Based on exact mass and fragmentation patterns, twenty phenolic compounds and their isomers were identified by MS: oleoside, hydroxytyrosol glucoside, hydroxtyrosol, elenolic acid glucoside, verbascoside, vanillin, demethyloleuropein, rutin, luteolin-*O*-glucoside, luteolin rutinoside, nuzhenide, caffeoyl-6-secologanoisde, oleuropein, oleuropein-aglycone di-aldehyde (3,4-DHPEA-EDA), oleuropein aglycone, oleuroside, oleocanthal (*p*-HPEA-EDA), and apigenin [44,45,46].The calibration plots indicate good correlations between peak areas and commercial standard concentrations. Regression coefficients were higher than 0.990. The limit of quantification (LOQ) was 8.3 µg/mL.

### 2.4. Modification of the Fe_3_O_4_ Particles and OMWW Treatment

The preparation of magnetic Fe_3_O_4_@C18 composite materials was carried out as follows. Dried magnetic Fe_3_O_4_ material (0.5 g) was added to a 100-mL three-necked bottle. Then, 25 mL purified toluene was added; this was followed by sonication for 1 h. After the Fe_3_O_4_ material had sedimented, the upper layer was decanted. Another 25 mL of purified toluene was added. Under N_2_, 0.5 mL of C18-SiCl_3_ was added dropwise. The reaction was continued with a stirring rate of 500 rpm at 50 °C for 5 h. The reaction mixture was then washed with toluene and separated under a magnetic field, and was dried under vacuum at 60 °C for 12 h [47].

Two types of SDS-modified magnetic particles, Fe_3_O_4_@SDS and Fe_3_O_4_@Al_2_O_3_@SDS particles, were prepared, so they could later be compared for their extraction efficiency. To prepare Fe_3_O_4_@SDS, aluminium isopropoxide (1.0 g) was dissolved in ethanol (60 mL) to form a homogeneous solution. Then, Fe_3_O_4_ NPs (0.5 g) were added to the above solution, under ultrasonification, for 5 min. Afterwards, a mixture of water and ethanol (1:5 *v*/*v*) was added dropwise to the above solution under vigorous stirring for 30 min. Finally, the obtained product was centrifuged and washed several times with ethanol, and dried in an oven at 300 °C for 3 h [48]. To coat Fe_3_O_4_ or Fe_3_O_4_@Al_2_O_3_ with SDS, the particles (0.5 g) were shaken for 24 h with 5 mL SDS solutions (0.01, 0.02, 0.04, 0.06, and 0.1 g/mL) in the presence of 15 mg NaCl at pH values of 2, 4, 6, and 8. After shaking, the supernatant was discarded and the particles were washed thoroughly, initially with tap water and finally with distilled water. Then, the material was dried at 60 °C for 24 h (modified from Adak et al. [23]).

To prepare Fe_3_O_4_@CA, 0.5 g of Fe_3_O_4_ and 5 g of citric acid were added to 10 mL water, and the temperature was raised to 90 °C under continuous stirring for 90 min [49].

For each of the four types of Fe_3_O_4_ particles (unmodified, Fe_3_O_4_@C18, Fe_3_O_4_@SDS, and Fe_3_O_4_@CA), 0.5 g of particles were added to 100 mL of OMWW. The solution was shaken for 15 min (200 rpm). The particles were collected at the side of the beaker with a neodynium magnet (size: 30 × 30 × 10 mm; magnetisation: N45), and the OMWW was decanted. Then, 10 mL of EtOH was added to the Fe_3_O_4_ particles. The EtOH was shaken for 5 min (200 rpm) to desorb the polyphenols from the particles. The particles were collected again with a neodynium magnet, and the EtOH was decanted. The polyphenol concentration was determined. The Fe_3_O_4_ particles were reused, as they were successfully regenerated. This procedure was repeated in 15 cycles for each of the four differently (un)modified magnetic particles that we had prepared. A scheme depicting the treatment of OMWW by removing polyphenols with Fe_3_O_4_ particles can be found in Figure 1.

### 2.5. Determination of the Total Phenol Content (TPC)

For a rapid assessment of whether Fe_3_O_4_@SDS or Fe_3_O_4_@Al_2_O_3_@SDS magnetic particles had better phenol extraction properties, the fast procedure used for the determination of the TPC was the Folin–Ciocalteu (FC) method. The ethanol-desorbed polyphenolic compounds were diluted to fit the gallic acid calibration curve (0–20 g/L). In total, 700 µL of standard or sample, 200 µL of FC reagent: H_2_O (1:3) and 100 µL of 1M Na_2_(CO_3_) buffer were added together and incubated for 2 h in the dark (at room temperature). Absorption spectra were measured with an Epoch Microplate Spectrophotometer. Spectrophotometric readings were collected at 765 nm.

## 3. Results

The goal of our research was to valorise OMWW by collecting polyphenolic compounds by adsorption on four different types of (un)modified Fe_3_O_4_ particles, and desorption in an alcoholic solution (EtOH). Further processing, clean up, or separation can consequently make OMWW a viable new source of polyphenolic compounds in the food, pharmaceutical, and cosmetic industries.

First, Fe_3_O_4_, Fe_3_O_4_@C18, Fe_3_O_4_@CA, and Fe_3_O_4_@SDS particles were synthesized, and their hydrodynamic diameter and zeta potential were characterized. The results are given in Table 1. As expected, the initial hydrodynamic diameter is in the upper end of the nano-sized range; therefore, easy agglomeration occurs, and the particles need to be dispersed by shaking. The advantage of agglomeration is that the particles are easily removed from the system.

### 3.1. Adsorption and Desorption of Polyphenols with the Unmodified Fe_3_O_4_

In the first experiment, the unmodified Fe_3_O_4_ magnetic particles were used to collect the polyphenolic compounds from OMWW. The concentrations of desorbed polyphenol were measured in EtOH (see Table 2). The polyphenol concentration in OMWW was measured by filtering OMWW through a syringe filter with a pore size of 0.2 µm. In this way, we separated the soluble portion of polyphenol (about 3–4 mg/L) from the insoluble portion of polyphenol in OMWW. Since a high proportion of the polyphenolic compounds is attached to the olive fruit particles in OMWW (leftover from the olive oil processing), the total polyphenol concentration in OMWW from Slovenian Istria can reach up to 27 mg/L [45].

The first polyphenol removal cycle with the unmodified Fe_3_O_4_ particles yielded what appeared to be a very low quantity of the targeted compounds (0.052 mg per mL of OMWW). However, the Fe_3_O_4_ particles were easily regenerated and reused, enabling a closed-loop process with several extraction cycles. Therefore, we tested a system where these particles were cycled fifteen times between the adsorption (OMWW) and desorption (EtOH) processes, with each repetition measured separately. The results are summarised in Table 1, where it can be clearly seen that, even after fifteen cycles, the Fe_3_O_4_ particles were still taking up polyphenolic compounds, proving their reusability. Most polyphenolic compounds are adsorbed by and desorbed from the particles in similar concentrations in each cycle, even after fifteen cycles. The main exceptions are verbascoside isomers, luteolin-*O*-glucoside isomers, oleuropein/oleuroside, oleuropein aglycone isomers, and apigenin, for which the measured concentration in the desorbed samples substantially decreased after 15 uptake cycles (Table 1, column 3). As the amount of polyphenols collected is about 0.05 mg per mL of OMWW, we should expect a maximum decrease in the polyphenol concentration in OMWW of about 0.75 mg/mL. In reality, a much higher decrease in polyphenol concentration is observed, from 3.44 to 1.85 mg/mL—Table 1, column 4 and 5. This means that the treatment also leads to the partial degradation of the polyphenolic compounds. This conclusion can also be made in relation to compounds such as oleuropein/oleoroside and oleuropein aglycone isomers, hydroxytyrosol, and demethyloleuropein; the concentrations of these compounds in OMWW after the 15 cycles (Table 1, column 5) decrease much more than the concentrations of the polyphenols that are collected by Fe_3_O_4_. On the other hand, the formation of different polyphenolic compounds, such as *p*-HPEA-EDA and nuzhenide isomers, can be observed in OMWW. Unmodified Fe_3_O_4_ particles do not only collect soluble polyphenols, but also polyphenolic compounds that are attached to the remaining olive particles. This can be concluded from the observation that compounds such as verbascoside and luteolin-*O*-glucoside isomers, luteolin rutinoside, and apigenin are not present in the soluble OMWW fraction (Table 1, column 3), but are detected in the ethanol fraction. Additionally, β-OH-verbascoside isomers are found in the ethanol fraction, but no decrease in the soluble β-OH-verbascoside content of OMWW was detected.

### 3.2. Adsorption and Desorption of Polyphenols with Fe_3_O_4_ Particles Modified with C18 Silica Gel

In the second set of experiments, Fe_3_O_4_ particles modified with C18 silica gel (Fe_3_O_4_@C18) were used. Compared to the adsorption and desorption with unmodified Fe_3_O_4_, a single removal cycle yielded a slightly higher, but still very low quantity of the targeted compounds (0.06 mg/mL of OMWW). The results are summarised in Table 3, where it can be clearly seen that, even after fifteen cycles, the modified Fe_3_O_4_ particles are still taking up polyphenolic compounds, proving their reusability. Most polyphenolic compounds are adsorbed by and desorbed from the particles in similar concentrations, even after fifteen cycles. The main exceptions are luteolin-*O*-glucoside and oleuropein aglycone isomers, hydroxytyrosol, and apignenin, for which the concentration substantially decreased after 15 uptake cycles. As the amount of polyphenols collected is about 0.06 mg per mL of OMWW per cycle, we should expect a maximum decrease in the polyphenol concentration in OMWW of about 0.9 mg/mL. In reality, a much higher decrease in polyphenol concentration was observed (from 3.02 to 1.63 mg/mL). This means that this treatment, like the treatment with unmodified Fe_3_O_4_ particles, also leads to the partial degradation of the polyphenolic compounds. This conclusion can also be made in relation to compounds such as hydroxytyrosol, trans *p*-coumaric acid 4-glucoside, caffeic acid and demethyloleuropein; the concentrations of these compounds in OMWW decrease much more than those of the polyphenols that are collected by Fe_3_O_4_. On the other hand, different polyphenolic compounds (β-OH-verbascoside and oleuropein aglycone isomers, vanillin, and *p*-HPEA-EDA) were observed in OMWW after the 15 removal cycles in higher quantities than in the initial OMWW. They could have been released from the remaining olive particles, or as a result of the breakdown of a larger compound. Fe_3_O_4_@C18 particles do not only collect soluble polyphenols, but also polyphenolic compounds attached to olive particles. This can be concluded from the observation that compounds such as rutin, luteolin-*O*-glucoside and verbascoside isomers, vanillin, and apigenin are not present in the soluble OMWW fraction, but are detected in the ethanol fraction desorbed from the modified magnetic particles. Another phenomenon that supports this conclusion is that hydroxytyrosol glucoside and caffeoyl-6-secologanoside were found in the desorbed ethanol fractions, but no decrease in the content of soluble hydroxytyrosol glucoside and caffeoyl-6-secologanoside in OMWW was detected.

### 3.3. Adsorption and Desorption of Polyphenols with Fe_3_O_4_ Particles Modified with Citric Acid

In the third set of experiments, Fe_3_O_4_ particles modified with citric acid (Fe_3_O_4_@CA) were used. With Fe_3_O_4_@CA, the amount of polyphenolic compounds collected per removal cycle (about 0.1 mg/mL) was double that collected by unmodified Fe_3_O_4_. The results are summarised in Table 4, where it can be clearly seen that, even after fifteen cycles, the CA-modified Fe_3_O_4_ particles are still taking up polyphenolic compounds, proving their reusability. In contrast to unmodified Fe_3_O_4_ and Fe_3_O_4_@C18, where the composition of collected polyphenolic compounds in the ethanolic fraction is fairly similar, Fe_3_O_4_@CA’s adsorption and desorption of polyphenols led to slight changes over the course of the 15 removal cycles. The concentration of compounds such as oleoside, elenolic acid glucoside, luteolin-*O*-glucoside and oleuropein/oleuroside isomers, and caffeic acid, trans *p*-coumaric acid 4-glucoside, and apigenin substantially decreased after 15 uptake cycles. Conversely, an increase in hydroxytyrosol and verbascoside isomers was observed. As the amount of polyphenols collected is about 0.1 mg per mL of OMWW, we should expect a maximum decrease in the polyphenol concentration in OMWW of about 1.5 mg/mL. The soluble phenolic content in OMWW decreased from 3.56 to 2.84 mg/mL, which is within the expected range. This means that Fe_3_O_4_@CA is a gentler removal method than unmodified Fe_3_O_4_ and Fe_3_O_4_@C18, leading to no or minimal degradation. This conclusion can also be supported by the observation that no compounds present in OMWW after treatment decrease in concentration much more than those that were collected in ethanol by Fe_3_O_4_@CA. On the other hand, an increase in the quantity of different polyphenolic compounds, such as hydroxytyrosol, trans *p*-coumaric acid 4-glucoside, caffeic acid, β-OH-verbascoside isomers, demethyloleuropein, luteolin-*O*-glucoside isomers, nuzhenide isomers, and caffeoyl-6-secologanoside, was observed in OMWW after the treatment. These compounds are most likely released from organic matter during the removal process under the influence of citric acid. Fe_3_O_4_@C18 particles do not only collect soluble polyphenols, but also polyphenolic compounds attached to particles. This can be concluded from the observation that compounds such as trans *p*-coumaric acid 4-glucoside, luteolin rutinoside, nuzhenide isomers, 3,4-DHPEA-EDA, oleuropein aglycone isomers, and apigenin were not present in the soluble OMWW fraction, but are detected in the ethanol fraction, having been desorbed from the modified magnetic particles. Another observation supporting this conclusion is that oleuropein/oleuroside isomers and *p*-HPEA-EDA were found in the ethanol fraction, but no substantial decrease in the soluble content in OMWW was detected.

### 3.4. Adsorption and Desorption of Polyphenols with Fe_3_O_4_ Particles Modified with SDS, Both with and without an Al_2_O_3_ Coating

In the fourth set of experiments, SDS-modified Fe_3_O_4_ particles with and without an Al_2_O_3_ coating (Fe_3_O_4_@SDS and Fe_3_O_4_@Al_2_O_3_@SDS, respectively) were used. First, different parameters for the synthesis of the particles were tested (pH of synthesis solution, presence of Al_2_O_3_ coating, SDS concentration) to obtain the highest polyphenol removal efficiency in OMWW. In this experiment, the total phenol concentrations desorbed in ethanol were determined using the fast spectrophotometric Folin–Ciocalteu method. The results can be found in Table 5. In general, Fe_3_O_4_@SDS magnetic particles had a better removal efficiency than Fe_3_O_4_@Al_2_O_3_@SDS. From the results, it can be seen that, when keeping the SDS concentration constantand adding the Fe_3_O_4_ particles to an aqueous solution with 0.1 g/mL SDS, a constant pH of 5.5 during the particle modification proved to be the optimal parameters for the modification procedure.

The SDS-modified magnetic particles proved more effective than unmodified Fe_3_O_4_ and Fe_3_O_4_@C18, but not as effective as Fe_3_O_4_@CA. The results are summarised in Table 6, where it can be clearly seen that, even after fifteen cycles, the Fe_3_O_4_@SDS particles were still taking up polyphenolic compounds, proving their reusability. An interesting phenomenon here was that the uptake efficiency of certain compounds increased over the 15 cycles: this was the case for compounds such as oleoside isomers, hydroxytyrosol glucoside, and vanillin. As the amount of polyphenols collected was about 0.07 mg per mL of OMWW, we should expect a maximum decrease in the polyphenol concentration in OMWW of about 1.1 mg/mL. The soluble phenolic content in OMWW decreased from 3.85 to 2.57 mg/mL, which is within the expected range. This means that Fe_3_O_4_@SDS is more gentle removal method than unmodified Fe_3_O_4_ and Fe_3_O_4_@C18, leading to no or minimal degradation. This conclusion is also supported by the observation that no compounds in OMWW decrease in concentration much more than when they are collected in ethanol by Fe_3_O_4_@SDS. On the other hand, the formation of different polyphenolic compounds, such as oleoside isomers, hydroxytyrosol, demethyloleuropein, luteolin-*O*-glucoside, and apigenin, can be observed in OMWW. From the results in this study, it can be concluded that Fe_3_O_4_@SDS particles do not only collect soluble polyphenols, but also polyphenolic compounds attached to particles, because compounds such as vanillin, verbascoside isomers, luteolin-*O*-glucoside, luteolin rutinoside, nuzhenide, 3,4-DHPEA-EDA isomers, oleuropein aglycone isomers, and apigenin were not present in the soluble OMWW fraction, but were detected in the ethanol fraction. Another phenomenon confirming this statement is that hydroxytyrosol glucoside and caffeoyl-6-secologanoside were found in the ethanol fraction, but no decrease in the soluble hydroxytyrosol glucoside and caffeoyl-6-secologanoside content in OMWW was detected.

## 4. Discussion

Conventional adsorption beds have the capacity to effectively remove phenol compounds from OMWW. However, their regeneration requires either thermal or chemical methods, which increases the cost of the procedure and can have undesired environmental effects. Therefore, this work tested the possibility of using (un)modified Fe_3_O_4_ particles, which can be magnetically collected; this type of polyphenol collection possesses important traits, such as affordability, regeneration and reusability, and the non-hazardous disposal of spent adsorbent.

The advantage of our procedure, compared to molecularly imprinted polymers, is that a mixture of polyphenolic compounds can be collected. This is useful for specific applications when several compounds or groups of compounds are wanted, such as in food supplements, where mixtures of compounds can have synergistic beneficial effects. It can also be a good starting point for the subsequent chromatographic separation of polyphenolic compounds, since the compounds are present in a less complex matrix. Chromatographic separation may be a simpler and faster technique for the separation of compounds from complex mixtures, compared to finding an imprinted polymer for each separate polyphenolic compound. Moreover, in contrast to former studies, our study tested the removal efficiency of several polyphenolic compounds. The silica-coated magnetic nanoparticles have previously only been tested in the extraction of xanthohumol in beer [33]. The 1-hexadecyl-3-methylimidazolium bromide-coated Fe_3_O_4_ magnetic nanoparticles have only been tested in the collection of 2,4-dichlorophenol and 2,4,6-trichlorophenol from environmental water samples [34]. Finally, the use of Fe3O_4_ is a more sensible choice for industrial applications [50] than the use of carbon nanotubes [35], which are difficult to work with and expensive [51], or gold nanoparticles [40], which have been used in previous studies to collect polyphenolic compounds.

Removing polyphenolic compounds from OMWW via (un)modified Fe_3_O_4_ particles proved to be a promising technique when a multi-step approach was used, by repeating several cycles in which polyphenols were adsorbed onto the particles and then desorbed into a solvent. This technique is economically profitable in a system where the Fe_3_O_4_ particles can start a new cycle after desorption, and the solvent can be reused by evaporation, leading to the concentration of the polyphenolic compounds in small solvent volumes.

Our experimental results show that (un)modified Fe_3_O_4_ adsorbs free polyphenolic compounds, as well as polyphenolic compounds which are attached to particulate matter. It was also noted that unmodified Fe_3_O_4_ particles and Fe_3_O_4_@C18 cause some polyphenol degradation in OMWW, while Fe_3_O_4_@CA releases polyphenolic compounds from olive particulate matter inside the OMWW. Different modifications lead to different adsorption behaviours for each polyphenolic compound. This is due to different interactions between the polyphenolic compound and the magnetic particles.

The removal of polyphenolic compounds from OMWW with bare Fe_3_O_4_ attraction is mainly controlled by chemisorption combined with π-π interactions, along with water-bridged H-bonds, according to Dehmani et al. [52], or physiosorption, according to Yoon et al. [53]. Coating the Fe_3_O_4_ particles can lead to better efficiency in removing polyphenolic compounds from aqueous solutions [52]; this can also be seen in our results.

The adsorption of polyphenols from aqueous solution by C18 silica-gel-modified magnetic particles is the result of apolar Van der Waals forces. Therefore, differences in the polarity and solubility of the phenols between the aqueous and the solid apolar phases causes the mass transfer [54]. For this reason, the efficiency of the sorbent is related to the hydrophobicity of the compound [54]. In Table 2, this is represented by a higher uptake of more apolar compounds (i.e., those that eluted from the apolar chromatographic column at higher retention times) than in other treatments.

Fe_3_O_4_ magnetic particles modified with citric acid retain a high capacity to adsorb less hydrophilic compounds, and gain the ability to interact with polar molecules due to stronger interactions, including dipole–dipole or hydrogen interactions [54]. Additionally, a polar surface is more wettable, and consequently supports mass transfer of the more polar species from the aqueous solution to the sorbent [54]. This can be seen in our results in Table 3, in the fact that Fe_3_O_4_@CA magnetic particles favour the uptake of the earlier-eluting polar compounds, especially oleoside and caffeic acid. In addition to this, our results show that citric acid releases polyphenolic compounds from the organic matter into the aqueous phase of OMWW. This phenomenon is in accordance with the reports that organic acids might weaken or disintegrate cell membranes, simultaneously dissolving the polyphenolics and stabilizing them [55].

SDS-modified magnetic particles were initially expected by the authors to preferentially bind water-insoluble molecules, because a single layer of the SDS molecules on the surface of metallic particles is normally oriented in such a way that its apolar chains are exposed to the aqueous environment. From the results of Table 5, however, it can be seen that the adsorption and desorption of more polar phenols (faster eluting from the chromatography column) are favoured. This can be explained by the fact that SDS molecules can also form admicelles on metal oxide particles, leaving the polar group of SDS exposed to its surroundings [56].

Since OMWW consists of phenols with significantly different properties, one sorbent may be unable to collect all of the compounds in sufficient or desired quantities. Therefore, to obtain large enough quantities of these compounds, while also retaining selectivity for all analytes, a combination of the modified Fe_3_O_4_ compounds would be an obvious solution. For example, if several adsorption–desorption cycles with Fe_3_O_4_@CA magnetic particles were combined with subsequent cycles using Fe_3_O_4_@C18 magnetic particles, both polar and apolar compounds would be collected. For example, if vanillin were a polyphenolic compound of particular interest, a combination of Fe_3_O_4_@SDS magnetic particles could be added.

## 5. Conclusions

In this study, it was found that the major advantage of (un)modified Fe_3_O_4_ particles is their easy multiple-cycle regeneration using low concentrations of low-cost chemicals.Their demonstrated adsorption capacity has the potential for successful commercialization in industrial applications.Differently modified Fe_3_O_4_ particles exhibit different extraction efficiencies for polyphenols with different chemical and physical properties.A sequential extraction by differently modified particles offers the possibility of either a “complete extraction” of all polyphenols in the desired quantities, or a more targeted extraction of select molecules.

## Figures and Tables

**Figure 1 nanomaterials-12-02327-f001:**
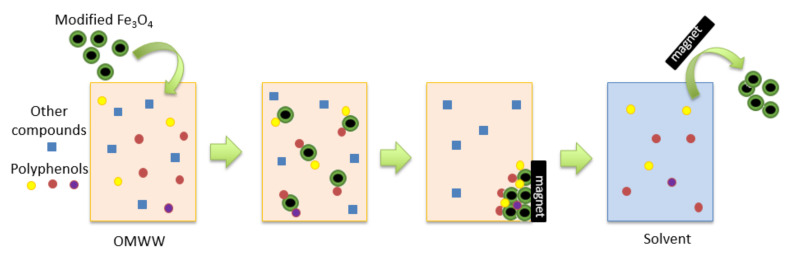
Scheme depicting the removal of polyphenols from olive mill waste water (OMWW) by the use of iron oxide (Fe_3_O_4_) particles and desorption in a solvent. Polyphenolic compounds are indicated by circular shapes in various colours, other compounds are represented by squares.

**Table 1 nanomaterials-12-02327-t001:** Hydrodynamic diameter and zeta potential of synthesized iron oxide (Fe_3_O_4_), iron oxide modified with silica gel (Fe_3_O_4_@C18), iron oxide modified with citric acid (Fe_3_O_4_@CA), and iron oxide modified with sodium dodecyl sulphate (Fe_3_O_4_@SDS) particles.

Particle Type	Hydrodynamic Diameter (nm)	Zeta Potential (mV)
Fe_3_O_4_	247.7	16.53
Fe_3_O_4_@C18	324.6	9.21
Fe_3_O_4_@CA	778.6	−36.40
Fe_3_O_4_@SDS	325.0	1.86

**Table 2 nanomaterials-12-02327-t002:** Polyphenol quantities for fifteen subsequent treatments of olive mill waste water (OMWW) with unmodified Fe_3_O_4_ particles. The particles were thereafter desorbed in ethanol (EtOH). The total concentration of polyphenolic compounds is quantified in mg per mL of OMWW; individual compounds are semi-quantified (using counts from the mass spectrometer (MS) detector).

Phenolic Compounds	Polyphenol Content in First EtOH Fraction	Polyphenol Content in Fifteenth EtOH Fraction	Soluble Polyphenol Content in OMWW—Before Treatment	Soluble Polyphenol Concentration in OMWW—After Treatment
Oleoside isomers	8565 ± 690	9838 ± 120	728,146 ± 37,843	660,565 ± 15,705
Hydroxytyrosol glucoside	6713 ± 768	6338 ± 204	62,868 ± 8439	37,418 ± 5137
Hydroxytyrosol	9271 ± 744	8879 ± 664	379,040 ± 24,353	108,364 ± 10,466
Trans *p*-coumaric acid 4-glucoside	698 ± 75	537 ± 52	29,560 ± 92	34,050 ± 990
Caffeic acid	12,527 ± 307	12,549 ± 1414	285,428 ± 8107	134,697 ± 19,500
Elenolic acid glucoside isomers	530 ± 14	706 ± 129	25,333 ± 1073	26,746 ± 318
β-OH-verbascoside isomers	5658 ± 1140	5338 ± 122	136,176 ± 438	136,466 ± 6029
Vanilin	<LOD	<LOD	<LOD	<LOD
Verbascoside isomers	7533 ± 388	1350 ± 118	<LOD	<LOD
Demethyloleuropein	176 ± 68	103 ± 8	14,483 ± 38	2752 ± 1993
Rutin	<LOD	<LOD	<LOD	<LOD
Luteolin-O-glucoside isomers	404 ± 107	224 ± 87	<LOD	<LOD
Luteolin rutinoside	573 ± 97	404 ± 111	<LOD	<LOD
Nuzhenide Isomers	116 ± 40	123 ± 11	1361 ± 64	5331 ± 315
Caffeoyl-6-secologanoside	5840 ± 349	5569 ± 360	129,754 ± 7106	112,055 ± 14,199
3,4-DHPEA-EDA isomers	174 ± 30	118 ± 29	4149 ± 743	2363 ± 1929
Oleuropein/Oleuroside isomers	428 ± 86	240 ± 49	23,635 ± 1135	12,346 ± 1231
Oleuropein aglycone Isomers	567 ± 113	241 ± 29	8297 ± 99	3815 ± 363
*p*-HPEA-EDA	155 ± 33	161 ± 7	<LOD	2758 ± 166
Apigenin	722 ± 38	435 ± 106	<LOD	<LOD
**Total (mg/mL)**	**0.052 ± 0.010**	**0.044 ± 0.002**	**3.44 ± 0.09**	**1.85 ± 0.07**

**Table 3 nanomaterials-12-02327-t003:** Polyphenol quantities for fifteen subsequent treatments of OMWW with Fe_3_O_4_ particles modified with C18 silica gel. The particles were thereafter desorbed in EtOH. Total concentrations of polyphenolic compounds are quantified in mg per mL of OMWW; individual compounds are semi-quantified (using counts from the MS detector).

Phenolic Compounds	Polyphenol Content in First EtOH Fraction	Polyphenol Content in Fifteenth EtOH Fraction	Soluble Polyphenol Content in OMWW—Before Treatment	Soluble Polyphenol Concentration in OMWW—After Treatment
Oleoside isomers	12,483 ± 219	14,202 ± 52	429,192 ± 48,396	402,504 ± 878
Hydroxytyrosol glucoside	8214 ± 462	8610 ± 210	23,162 ± 6222	20,171 ± 1453
Hydroxytyrosol	6059 ± 86	3805 ± 500	284,414 ± 3023	35,406 ± 6058
Trans *p*-coumaric acid 4-glucoside	483 ± 61	566 ± 51	23,900 ± 5967	<LOD
Caffeic acid	6791 ± 1135	5307 ± 344	363,362 ± 42,318	54,073 ± 4322
Elenolic acid glucoside isomers	845 ± 94	800 ± 8	38,053 ± 2452	20,489 ± 231
β-OH-verbascoside isomers	10,514 ± 851	10,198 ± 663	16,641 ± 4222	138,969 ± 17,363
Vanilin	799 ± 133	791 ± 6	<LOD	27,458 ± 1018
Verbascoside isomers	10,743 ± 65	12,817 ± 476	<LOD	<LOD
Demethyloleuropein	429 ± 61	324 ± 68	19,732 ± 217	3256 ± 40
Rutin	2206 ± 255	1877 ± 81	<LOD	<LOD
Luteolin-O-glucoside isomers	1278 ± 83	769 ± 41	<LOD	<LOD
Luteolin rutinoside	<LOD	<LOD	<LOD	<LOD
Nuzhenide Isomers	158 ± 14	159 ± 5	6662 ± 429	5038 ± 116
Caffeoyl-6-secologanoside	10,291 ± 1405	9053 ± 802	117,566 ± 8118	119,630 ± 4237
3,4-DHPEA-EDA isomers	15,507 ± 1088	12,357 ± 122	<LOD	29,595 ± 197
Oleuropein/Oleuroside	1131 ± 296	965 ± 13	26,995 ± 1438	17,485 ± 205
Oleuropein aglycone Isomers	2527 ± 775	854 ± 13	3378 ± 2605	19,622 ± 3150
*p*-HPEA-EDA	2360 ± 182	2332 ± 9	12,601 ± 85	17,819 ± 264
Apigenin	839 ± 160	293 ± 4	<LOD	<LOD
**Total (mg/mL)**	**0.064 ± 0.005**	**0.058 ± 0.001**	**3.02 ± 0.12**	**1.63 ± 0.18**

**Table 4 nanomaterials-12-02327-t004:** Polyphenol quantities for fifteen subsequent treatments of OMWW with Fe_3_O_4_ particles modified with citric acid. The particles were thereafter desorbed in EtOH. Total concentrations of polyphenolic compounds are quantified in mg per mL of OMWW; individual compounds are semi-quantified (using counts from the MS detector).

Phenolic Compounds	Polyphenol Content in First EtOH Fraction	Polyphenol Content in Fifteenth EtOH Fraction	Soluble Polyphenol Content in OMWW—Before Treatment	Soluble Polyphenol Concentration in OMWW—After Treatment
Oleoside isomers	154,399 ± 12,489	129,203 ± 4337	2308,995 ± 371,959	1506,070 ± 82,930
Hydroxytyrosol glucoside	15,281 ± 489	18,323 ± 2255	36,111 ± 36	173,218 ± 14,866
Hydroxytyrosol	11,419 ± 1154	28,553 ± 1403	124,526 ± 16,804	105,736 ± 26,879
Trans *p*-coumaric acid 4-glucoside	1138 ± 122	<LOD	<LOD	33,190 ± 5006
Caffeic acid	57,636 ± 2078	44,890 ± 5083	63,215 ± 14,291	438,410 ± 85,673
Elenolic acid glucoside isomers	11,889 ± 122	5144 ± 2582	152,095 ± 11,816	88,787 ± 21,987
β-OH-verbascoside isomers	13,574 ± 1434	14,020 ± 44	193,562 ± 77,384	520,700 ± 4087
Vanilin	<LOD	<LOD	<LOD	<LOD
Verbascoside isomers	1246 ± 44	2215 ± 205	<LOD	<LOD
Demethyloleuropein	223 ± 69	225 ± 16	1970 ± 583	3778 ± 40
Rutin	<LOD	<LOD	<LOD	<LOD
Luteolin-O-glucoside isomers	2800 ± 100	1814 ± 138	9843 ± 3992	18,080 ± 393
Luteolin rutinoside	1537 ± 420	1105 ± 90	<LOD	<LOD
Nuzhenide Isomers	128 ± 16	110 ± 35	<LOD	4661 ± 1752
Caffeoyl-6-secologanoside	12,005 ± 1320	9786 ± 234	118,057 ± 8281	266,715 ± 4646
3,4-DHPEA-EDA	61 ± 26	210 ± 9	<LOD	<LOD
Oleuropein/Oleuroside isomers	1211 ± 279	456 ± 155	11,488 ± 2688	12,531 ± 3935
Oleuropein aglycone Isomers	341 ± 34	<LOD	<LOD	<LOD
*p*-HPEA-EDA	245 ± 1	126 ± 5	3781 ± 245	3317 ± 385
Apigenin	2426 ± 23	1391 ± 100	<LOD	<LOD
**Total (mg/mL)**	**0.100 ± 0.009**	**0.095 ± 0.020**	**3.56 ± 0.18**	**2.84 ± 0.15**

**Table 5 nanomaterials-12-02327-t005:** Treatment of OMWW with Fe_3_O_4_@SDS or Fe_3_O_4_@Al_2_O_3_@SDS with different synthesis parameters (pH, SDS concentration). The concentration of the desorbed total phenol content was measured using the Folin–Ciocalteu method. The standard deviation of the results is 0.02 mg per mL OMWW.

Particle Type	Concentration SDS (g/mL)	pH	Total Phenol Concentration in EtOH (mg per mL OMWW in GAE)
Fe_3_O_4_@Al_2_O_3_	0.01	4.5	0.11
Fe_3_O_4_@Al_2_O_3_	0.02	4.5	0.13
Fe_3_O_4_@Al_2_O_3_	0.05	4.5	0.17
Fe_3_O_4_@Al_2_O_3_	0.1	4.5	0.20
Fe_3_O_4_@Al_2_O_3_	0.02	3.5	0.17
Fe_3_O_4_@Al_2_O_3_	0.02	4.5	0.11
Fe_3_O_4_@Al_2_O_3_	0.02	5.5	0.16
Fe_3_O_4_@Al_2_O_3_	0.02	8	0.09
Fe_3_O_4_	0.01	4.5	0.25
Fe_3_O_4_	0.02	4.5	0.19
Fe_3_O_4_	0.05	4.5	0.10
Fe_3_O_4_	0.1	4.5	0.28
Fe_3_O_4_	0.02	3.5	0.14
Fe_3_O_4_	0.02	4.5	0.20
Fe_3_O_4_	0.02	5.5	0.37
Fe_3_O_4_	0.02	8	0.32

**Table 6 nanomaterials-12-02327-t006:** Polyphenol quantities for fifteen subsequent treatments of OMWW with Fe_3_O_4_ particles modified with sodium dodecyl sulphate. The particles were thereafter desorbed in EtOH. Total concentrations of polyphenolic compounds are quantified in mg per mL of OMWW; individual compounds are semi-quantified (using counts from the MS detector).

Phenolic Compounds	Polyphenol Content in First EtOH Fraction	Polyphenol Content in Fifteenth EtOH Fraction	Soluble Polyphenol Content in OMWW—Before Treatment	Soluble Polyphenol Concentration in OMWW—After Treatment
Oleoside isomers	54,046 ± 7914	73,216 ± 15,453	898,000 ± 1373	991,238 ± 71,747
Hydroxytyrosol glucoside	937 ± 319	11,109 ± 1372	75,035 ± 9793	69,315 ± 2171
Hydroxytyrosol	18,578 ± 3650	12,386 ± 1727	44,488 ± 16,232	98,379 ± 8494
Trans *p*-coumaric acid 4-glucoside	613 ± 109	<LOD	21,878 ± 3758	15,080 ± 1676
Caffeic acid	26,446 ± 1207	22,178 ± 132	86,608 ± 746	15,864 ± 6493
Elenolic acid glucoside isomers	11,965 ± 145	11,797 ± 330	72,928 ± 17,864	38,435 ± 8553
β-OH-verbascoside isomers	11,911 ± 884	14,883 ± 106	230,313 ± 9139	224,923 ± 19,315
Vanilin	2377 ± 141	7850 ± 3802	<LOD	<LOD
Verbascoside isomers	7818 ± 510	2310 ± 505	<LOD	<LOD
Demethyloleuropein	151 ± 21	179 ± 2	1853 ± 21	4127 ± 39
Rutin	<LOD	<LOD	<LOD	<LOD
Luteolin-O-glucoside	2124 ± 6	2101 ± 113	<LOD	56,390 ± 2592
Luteolin rutinoside	1100 ± 93	<LOD	<LOD	<LOD
Nuzhenide Isomers	82 ± 5	123 ± 23	<LOD	<LOD
Caffeoyl-6-secologanoside	7390 ± 280	8055 ± 923	139,568 ± 15,822	129,294 ± 3056
3,4-DHPEA-EDA isomers	94 ± 15	144 ± 6	<LOD	<LOD
Oleuropein/Oleuroside isomers	308 ± 43	555 ± 174	25,820 ± 7122	27,072 ± 7686
Oleuropein aglycone Isomers	286 ± 5	215 ± 9	<LOD	<LOD
*p*-HPEA-EDA	139 ± 73	131 ± 6	4253 ± 43	2389 ± 434
Apigenin	1026 ± 49	1240 ± 59	<LOD	2371 ± 383
**Total (mg/mL)**	**0.071 ± 0.009**	**0.083 ± 0.03**	**3.85 ± 0.14**	**2.57 ± 0.05**

## Data Availability

Data are contained within the article.

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
