# Peer review of "The Use of Modified Fe3O4 Particles to Recover Polyphenolic Compounds for the Valorisation of Olive Mill Wastewater from Slovenian Istria"

_nanomaterials, 2022, doi:10.3390/nano12142327_

Round 1
Reviewer 1 Report
Very nice article, clearly written and comprehensive. So I recommend its acceptance.
Just a very few (and little) points that could be considered to achieve an even larger clarity of the text.
abstract l 18 (and similar later in the text). I think the usage of the phrase " (un)modified iron oxide magnetic particles (Fe3O4, Fe3O4@C18, etc." is confusing. Please consider a more precise wording here. In the same way, the magnetic particles are always of the same type, but you use 4 different ways of surface modification: unmodified, @C18, @CA, etc. as written in l152, I would not speak here from a magnetic particle type preparation (that I would understand as a different size/magentic moment/structure or so on). It then makes clearer how extremely important your surface modification for the capture of polyphenolic compounds is!
l150: you mention the acidification of OMWW, why is this so important here, maybe provide a short explanation?
l208: (Again I would denote here each of the 4 Fe3O4 systems or modifcation) but what do you mean by "a total of 5 g/L (which is a concentration right?) to 100 mL (a volume)" Is this right?
l210: You provide for the Magnet a strength of 343 N, what is meant here exactly, isn't the strength depending on the distance to the magnet? Please provide little more details.
figure 1 and ll215: In the procedure, you use the magnet to collect the particles with the polyphenols absorbed onto them from the OMWW. Then you desolve by adding OMWW, are during that time/step the particles kept at the same postion with the magnet? If you say the alcoholic solvent (the EtOH can be reused, how are the polyphenols removed from EtOH? I have the impression that the procedure shown figure 1 is not completely matching the description of the text
l323: is the plural in ...adsorption and desorption give"s" slight changes right?
I am a little missing in the discussion the comparison of the polyphenol gain and the performance of your Fe3O4 extraction procedure with the other established methods named in the introduction. Can't you give a short classification or estimation and maybe statement of advantages and disadvantages of your procedure in the discussion?
Author Response
Comment 1:
abstract l 18 (and similar later in the text). I think the usage of the phrase " (un)modified iron oxide magnetic particles (Fe3O4, Fe3O4@C18, etc." is confusing. Please consider a more precise wording here. In the same way, the magnetic particles are always of the same type, but you use 4 different ways of surface modification: unmodified, @C18, @CA, etc. as written in l152, I would not speak here from a magnetic particle type preparation (that I would understand as a different size/magentic moment/structure or so on). It then makes clearer how extremely important your surface modification for the capture of polyphenolic compounds is!
Answer to comment 1:
Line 18 was rewritten as follows: This research describes new ways of collecting polyphenolic compounds by using unmodified iron oxide (Fe3O4) particles and Fe3O4 modified with silica gel (Fe3O4@C18), citric acid (Fe3O4@CA) and sodium dodecyl sulphate (Fe3O4@SDS).
Line 152 was rewritten as follows: Fe3O4, Fe3O4@C18, Fe3O4@CA and Fe3O4@SDS were tested for their extraction efficiency, each at a different day.
Comment 2:
l150: you mention the acidification of OMWW, why is this so important here, maybe provide a short explanation?
Answer to comment 2:
Acidification is important because it can improve drastically the concentration of free polyphenolic compounds in OMWW. A sentence was added in the text: Acidic conditions can considerably increase the fraction of free phenolic compounds in OMWW. However, since these experiments were designed to collect polyphenolic compounds from OMWW on a large scale, OMWW was not acidified as recommended by Jerman Klen and Mozetič Vodpivec [44] because this would not be economically feasible.
Comment 3:
l208: (Again I would denote here each of the 4 Fe3O4 systems or modifcation) but what do you mean by "a total of 5 g/L (which is a concentration right?) to 100 mL (a volume)" Is this right?
Answer to comment 3:
Yes, it is correct. To avoid confusion, the sentence was changed as follows: For each of the four types of Fe3O4 particles (unmodified, Fe3O4@C18, Fe3O4@SDS, and Fe3O4@CA), 0.5 g of particles were added to 100 mL of OMWW.
Comment 4:
l210: You provide for the Magnet a strength of 343 N, what is meant here exactly, isn't the strength depending on the distance to the magnet? Please provide little more details.
Answer to comment 4:
Yes, the magnetic strength is dependent on the distance, I provided the strength given by the providers when there is no distance between the magnet and the object, as an indication of its strength (tables exist to calculate the actual magnetic strength then according to the distance towards an object). Since its seems to lead to confusion, the authors changed the magnetic description to its size (30x30x10 mm) and magnetisation (N45).
Comment 5:
figure 1 and ll215: In the procedure, you use the magnet to collect the particles with the polyphenols absorbed onto them from the OMWW. Then you desolve by adding OMWW, are during that time/step the particles kept at the same postion with the magnet? If you say the alcoholic solvent (the EtOH can be reused, how are the polyphenols removed from EtOH? I have the impression that the procedure shown figure 1 is not completely matching the description of the text
Answer to comment 5:
Thank you, this part of the sentence did indeed not make sense. The alcoholic solvent with polyphenols is our final product and is not reused. We corrected the sentence as follows: The Fe3O4 particles were reused, as they were successfully regenerated.
Comment 6:
l323: is the plural in ...adsorption and desorption give"s" slight changes right?
Answer to comment 6:
The spelling mistake was corrected.
Comment 7
I am a little missing in the discussion the comparison of the polyphenol gain and the performance of your Fe3O4 extraction procedure with the other established methods named in the introduction. Can't you give a short classification or estimation and maybe statement of advantages and disadvantages of your procedure in the discussion?
Answer to comment 7:
We thank the reviewer for this comment. The following text was added to the Discussion section of the manuscript:
The advantage of our procedure versus molecularly imprinted polymers is that a mixture of polyphenolic compounds can be collected. This is useful for specific applications when several compounds or groups of compounds are wanted such as in food supplements (mixtures of compounds can have synergistic beneficial effects). It also can be a good as a starting point for subsequent chromatographic separation of the polyphenolic compounds, since the compounds are present in a less complex matrix. Chromatographic separation may be a simpler and fast technique to get all compounds separated instead of finding for each separate polyphenolic compound an imprinted polymer. Second, in contrast to former studies, our study was testing the removal efficiency of several polyphenolic compounds. The silica-coated magnetic nanoparticles were only tested on extraction of xanthohumol in beer [33]. 1-hexadecyl-3-methylimidazolium bromide coated Fe3O4 magnetic nanoparticles were only tested on the collection of 2,4-dichlorophenol and 2,4,6-trichlorophenol from environmental water samples [34]. Third the use of Fe3O4 is a more reasonable choice for industrial applications [50] than the use of carbon nanotubes [35], which are difficult to work with and expensive [51] or gold nanoparticles [40], which were use in former studies to collect polyphenolic compounds.
Reviewer 2 Report
This paper deals with the treatment of olive oil wastewater for the recovery of polyphenols using magnetic iron oxides and modified magnetic iroxides. The well-studied recovery and reusability of various iron-oxides for poly phenols adsorptive removal from olive oil wastewater with HPLC methodology and their detection limits with the re-usability of more than 15 cycles were interesting. However, it is a lack of textural and size characteristics of the prepared materials. After improving the following aspects, it can be considered for publication.
1. Abstract stand alone in the manuscript, give a full form of the terms, Fe3O4, Fe3O4@C18, Fe3O4@CA, 18 Fe3O4@SDS used for the first time at abstract.
2. Novelty of this study should be explored by comparing it with previous literature on this topic in the introduction.
3. To know the size, surface area, and morphology of prepared material, it needs to study BET, SEM, and TEM analysis.
4. Moreover, to understand the Texture of the prepared materials need to analyze XRD studies.
Author Response
- Abstract stand alone in the manuscript, give a full form of the terms, Fe3O4, Fe3O4@C18, Fe3O4@CA, 18 Fe3O4@SDS used for the first time at abstract.
Answer to comment 1: The sentence was rewritten as follows: This research describes new ways of collecting polyphenolic compounds by using unmodified iron oxide (Fe3O4) particles and Fe3O4 modified with silica gel (Fe3O4@C18), citric acid (Fe3O4@CA) and sodium dodecyl sulphate (Fe3O4@SDS).
- Novelty of this study should be explored by comparing it with previous literature on this topic in the introduction.
Answer to comment 2: A paragraph was added in the discussion section: The advantage of our procedure molecularly imprinted polymers, is that a mixture of polyphenolic compounds can be collected. This is useful for specific applications when several compounds or groups of compounds are wanted such as in food supplements (mixtures of compounds can have synergistic beneficial effects). It also can be a good as a starting point for subsequent chromatographic separation of the polyphenolic compounds, since the compounds are present in a less complex matrix. Chromatographic separation may be a simpler and fast technique to get all compounds separated instead of finding for each separate polyphenolic compound an imprinted polymer. Second, in contrast to former studies, our study was testing the removal efficiency of several polyphenolic compounds. The silica-coated magnetic nanoparticles were only tested on extraction of xanthohumol in beer [33]. 1-hexadecyl-3-methylimidazolium bromide coated Fe3O4 magnetic nanoparticles were only tested on the collection of 2,4-dichlorophenol and 2,4,6-trichlorophenol from environmental water samples [34]. Third the use of Fe3O4 is a more reasonable choice for industrial applications [50] than the use of carbon nanotubes [35], which are difficult to work with and expensive [51] or gold nanoparticles [40], which were use in former studies to collect polyphenolic compounds.
- To know the size, surface area, and morphology of prepared material, it needs to study BET, SEM, and TEM analysis. 4. Moreover, to understand the Texture of the prepared materials need to analyze XRD studies.
Answer to comments 3 and 4: Since the methods to prepare the nanoparticles are previously described in literature, as well as the fact that different results on polyphenol collection are obtained with different modified particles proofs that Fe3O4 is coated properly. Understanding the structure of the particles seems to the author to not add lots of extra value to the goal of this paper. The authors did however some extra characterization of the particles by measuring the hydrodynamic diameter and zeta-potential as requested by reviewer 3:
First Fe3O4, Fe3O4@C18, Fe3O4@CA and Fe3O4@SDS particles were synthesized and their hydrodynamic diameter and zeta potential characterized. The results are given in Table 1. The results are as expected, the initial hydrodynamic diameter is in the higher nanosized range, therefore also easy agglomeration occurs and dispersion of the particles is needed by shaking. The advantage of agglomeration is that the particles are easily removed from the system in comparison with particles that stay dispersed.
Table 1. Hydrodynamic diameter and zeta potential of synthesized Fe3O4, Fe3O4@C18, Fe3O4@CA and Fe3O4@SDS particles
|
Particle type |
Hydrodynamic diameter (nm) |
Zeta potential (mV) |
|
Fe3O4 |
247.7 |
16.53 |
|
Fe3O4@C18 |
324.6 |
9.21 |
|
Fe3O4@CA |
778.6 |
-36.40 |
|
Fe3O4@SDS |
325.0 |
1.86 |
Reviewer 3 Report
The paper by Kelly Peeters et al. represents a way to recover polyphenolic compounds from olive mill wastewater, which is a large-scale by-product in the olive oil industry. In this regard the article has a reasonable and actual goal-setting with good mentioning of various methods utilized for this purpose and subsequent pros and cons delivered in the introduction section. Differently surface coated iron oxide nanoparticles, namely Fe3O4, Fe3O4@C18, Fe3O4@CA, Fe3O4@SDS (citric acid (CA), silica gel (C18) and sodium dodecyl sulphate (SDS)), were compared towards the efficiency of polyphenolic compounds adsorption/desorption and magnetic separation. Although this paper is well-written and pragmatically goal-oriented, I would recommend to strengthen particle characterization part. Dealing with the paper, the reader has no idea about structural parameters of the particles. At least the size (or hydrodynamic diameter) and zeta-potential should be given. Corresponding discussion with size(surface charge)-quantity of polyphenol is recommended for adding. The authors are encouraged to refer following papers (10.1016/j.colsurfa.2018.09.044; 10.1016/j.colsurfa.2013.12.009) as a distinct examples of coated iron oxide NPs.
Author Response
The paper by Kelly Peeters et al. represents a way to recover polyphenolic compounds from olive mill wastewater, which is a large-scale by-product in the olive oil industry. In this regard the article has a reasonable and actual goal-setting with good mentioning of various methods utilized for this purpose and subsequent pros and cons delivered in the introduction section. Differently surface coated iron oxide nanoparticles, namely Fe3O4, Fe3O4@C18, Fe3O4@CA, Fe3O4@SDS (citric acid (CA), silica gel (C18) and sodium dodecyl sulphate (SDS)), were compared towards the efficiency of polyphenolic compounds adsorption/desorption and magnetic separation. Although this paper is well-written and pragmatically goal-oriented, I would recommend to strengthen particle characterization part. Dealing with the paper, the reader has no idea about structural parameters of the particles. At least the size (or hydrodynamic diameter) and zeta-potential should be given. Corresponding discussion with size(surface charge)-quantity of polyphenol is recommended for adding. The authors are encouraged to refer following papers (10.1016/j.colsurfa.2018.09.044; 10.1016/j.colsurfa.2013.12.009) as a distinct examples of coated iron oxide NPs.
Answers to reviewer 3: The hydrodynamic diameter and zeta-potential were measured and added to the results in Table 1:
First Fe3O4, Fe3O4@C18, Fe3O4@CA and Fe3O4@SDS particles were synthesized and their hydrodynamic diameter and zeta potential characterized. The results are given in Table 1. The results are as expected, the initial hydrodynamic diameter is in the higher nanosized range, therefore also easy agglomeration occurs and dispersion of the particles is needed by shaking. The advantage of agglomeration is that the particles are easily removed from the system in comparison with particles that stay dispersed.
Table 1. Hydrodynamic diameter and zeta potential of synthesized Fe3O4, Fe3O4@C18, Fe3O4@CA and Fe3O4@SDS particles
|
Particle type |
Hydrodynamic diameter (nm) |
Zeta potential (mV) |
|
Fe3O4 |
247.7 |
16.53 |
|
Fe3O4@C18 |
324.6 |
9.21 |
|
Fe3O4@CA |
778.6 |
-36.40 |
|
Fe3O4@SDS |
325.0 |
1.86 |
The authors decided not to add the requested articles to the references and the Literature list, because they do not see any connection with olive mill wastewater treatment or Fe3O4 particle modification with SDS, C18 or CA.
Round 2
Reviewer 3 Report
Reviewer thanks the authors for swift revision. Nanomaterials is a highly reputed Q1 journal within the field of nano-. The papers accepted to the journal are high-quality and deep studies covering synthetic aspects of nanomaterials, comprehensive characterization by set of physical methods and eventually its application. From the point of weak synthetic part (all of iron oxides utilized in the work are known before) and lack of characterization of NPs (except of DLS) this submission is not within the scope of Nanomaterials.
For this the authors have to address https://www.mdpi.com/journal/nanomaterials/about. The scope is clearly defined as follows.
Nanomaterials are materials with typical size features in the lower nanometer size range and characteristic mesoscopic properties; for example quantum size effects. These properties make them attractive objects of fundamental research and potential new applications. The scope of Nanomaterials covers the preparation, characterization and application of all nanomaterials.
Due to the fact that particle sizes of 247-779 nm are not within the lower nanometer size range and are considered as microparticles, the studied objects are far away from the scope of the journal.
Second, in order to achieve the level of Nanomaterials typical methods as TEM, SEM, AFM, etc. before and after sorption of polyphenolic compounds should necessarily be exploited to support major findings of the article. The effect of nano- and evolution of nano- (aggregation, surface layer modifications) should be clearly delivered to the reader of Nanomaterials by microscopy images, diffraction patterns and other evidence.
Third, the research design is 100% same to previously published article in Molecules made for uncoated Fe3O4 particles (not nanoparticles). The authors should thoroughly explain what is new in this paper except only modificators of the surface. This is particularly necessary taking into account that “In this study it was found that the major advantage of (un)modified Fe3O4 particles is their easy multiple-cycle regeneration using low concentrations of low-cost chemicals.” That means that uncoated particles are better from this point. The authors have to pay special attention to discuss importance of surface modification ways of iron oxide NPs to justify goal-setting of the experimental design in a way better way. In this regard, suggested references (10.1016/j.colsurfa.2018.09.044; 10.1016/j.colsurfa.2013.12.009) will only strengthen introductory part of the article.
Fourth, the following result “Differently modified Fe3O4 particles exhibit different extraction efficiencies for pol-yphenols with different chemical and physical properties.” contains no useful information and should be rewritten or deleted. In order to have scientific meaning, the authors have to find regularities and correlations between structure and property or property1-property2. For this, electrokinetic potential or surface layer hydrophobicity should be correlated with pK or hydrophobic-hydrophilic balance of polyphenolic compounds.
Finally, there is no other recommendation can be given for this article than reject. Otherwise, since the Solid phase extraction and New separation methodologies are within the scope of MDPI Separations, this manuscript can be resubmitted or redirected to MDPI Separations.
